# Trametinib-Induced Epidermal Thinning Accelerates a Mouse Model of Junctional Epidermolysis Bullosa

**DOI:** 10.3390/biom13050740

**Published:** 2023-04-25

**Authors:** Grace Tartaglia, Pyung Hun Park, Michael H. Alexander, Alexander Nyström, Joel Rosenbloom, Andrew P. South

**Affiliations:** 1Department of Dermatology and Cutaneous Biology, Thomas Jefferson University, Philadelphia, PA 19107, USA; 2Department of Dermatology, Medical Faculty, Medical Center—University of Freiburg, 79104 Freiburg im Breisgau, Germany; 3The Joan and Joel Rosenbloom Research Center for Fibrotic Diseases, Thomas Jefferson University, Philadelphia, PA 19107, USA; 4Sidney Kimmel Cancer Center, Thomas Jefferson University, Philadelphia, PA 19107, USA

**Keywords:** junctional epidermolysis bullosa, Trametinib, inflammation, fibrosis, Losartan

## Abstract

Junctional epidermolysis bullosa (JEB) patients experience skin and epithelial fragility due to a pathological deficiency in genes associated with epidermal adhesion. Disease severity ranges from post-natal lethality to localized skin involvement with persistent blistering followed by granulation tissue formation and atrophic scarring. We evaluated the potential of utilizing Trametinib, an MEK inhibitor previously shown to target fibrosis, with and without the documented EB-anti-fibrotic Losartan for reducing disease severity in a mouse model of JEB; *Lamc2^jeb^* mice. We found that Trametinib treatment accelerated disease onset and decreased epidermal thickness, which was in large part ameliorated by Losartan treatment. Interestingly, a range of disease severity was observed in Trametinib-treated animals that tracked with epidermal thickness; those animals grouped with higher disease severity had thinner epidermis. To examine if the difference in severity was related to inflammation, we conducted immunohistochemistry for the immune cell markers CD3, CD4, CD8, and CD45 as well as the fibrotic marker αSMA in mouse ears. We used a positive pixel algorithm to analyze the resulting images and demonstrated that Trametinib caused a non-significant reduction in CD4 expression that inversely tracked with increased fibrotic severity. With the addition of Losartan to Trametinib, CD4 expression was similar to control. Together, these data suggest that Trametinib causes a reduction in both epidermal proliferation and immune cell infiltration/proliferation, with concurrent acceleration of skin fragility, while Losartan counteracts Trametinib’s adverse effects in a mouse model of JEB.

## 1. Introduction

Junctional epidermolysis bullosa (JEB) is a rare genetic blistering disease that results from a separation of the lamina lucida in the basement membrane of the skin after mechanical trauma [1,2]. Laminin-332 deficient JEB occurs as a result of a mutation in one of the three subunits of laminin 332: laminin α3 (LAMA3), laminin β3 (LAMB3), or laminin γ2 (LAMC2) [3]. Historically there have been two classifications of JEB that differ in severity, Herlitz and non-Herlitz. More recently, further sub-classifications of JEB have been determined based on severity and onset [4,5].

One viable mouse model of JEB is representative of the non-Herlitz form (*Lamc2^jeb^*), due to a hypomorphic allele of the Lamc2 gene causing reduced levels of laminin-332 [6]. The onset of the disease phenotype begins around 13–15 weeks of age, which includes ulcers and granulation tissue on their ears thought to be driven primarily through inflammation. It has been noted, however, that there is a sexual dimorphism among the *Lamc2^jeb^* mice in that males start showing disease phenotype before females do, but there is no noted difference in severity between the sexes [7]. The paws and tail can also experience epidermal separation with minimal inflammation. Since the *Lamc2^jeb^* mice experience substantial scarring over time, they are used as a model of EB-driven fibrosis.

Fibrosis is a progressive condition characterized by excessive deposition of extracellular matrix and an increase in activated fibroblasts, or myofibroblasts [8,9]. One common marker of myofibroblasts is alpha-smooth muscle actin, or αSMA. αSMA contributes to the increased contractility of myofibroblasts [10] to assist in wound healing and is often studied in the context of fibrosis to identify pathologic myofibroblasts. Inflammatory conditions during wounding contribute to the activation of fibroblasts, and the interdependent relationship between fibrosis and inflammation has been extensively researched but is still not fully understood [11]. The inability to restore damaged tissue integrity from unresolved wound healing leads to prolonged inflammation and an accumulation of fibrous connective tissue, which both contribute to fibrosis and potential cancer development [12,13]. Adaptive immunity plays a large role in fibrogenesis [14], particularly T cells as they regulate both pro-fibrotic and anti-fibrotic functions. One marker of infiltrating T cells that contributes to fibrosis is the cluster of differentiation 4 (CD4) [15,16]. Conventional CD4+ T cells recruit granulocytes to the site of infection and inflammation but also have an indirect role in fibroblast activation and extracellular matrix development and deposition by secreting inflammatory cytokines such as tumor necrosis factor- α and interleukin-1β [17,18].

T cell regulation is both disease and tissue-specific [19,20,21,22], making systematic investigations into their role in fibrosis imperative in developing effective treatments. One compound of widespread interest is Trametinib, an FDA-approved kinase inhibitor that targets mitogen-activated protein kinase (MEK), with published success as an anti-fibrotic [23]. Trametinib has also demonstrated anti-inflammatory effects in the brain upon injury [24], making it a promising avenue of investigation into a disease model featuring both inflammation and fibrosis. We also used Losartan, an angiotensin-II type 1 receptor blocker, with published success as an antifibrotic in the recessive dystrophic sub-type of epidermolysis bullosa both in murine studies and clinical trial [25,26,27] to determine if a combination regimen with Trametinib would provide a beneficial synergistic effect.

Ultimately, we sought to determine in a preliminary study whether Trametinib is a viable treatment to delay blister progression and reduce the severity of chronic inflammation or fibrosis development. We found that not only does Trametinib have a wide range in response in the JEB mouse model, but also a poor reaction to treatment caused a thinner epidermis and a decrease in CD4 expression that inversely correlated with an increase in αSMA. The combination of Trametinib and Losartan also mitigated the blistering phenotype across all mice, suggesting that Losartan is able to counteract Trametinib’s adverse effects in this murine model of JEB.

## 2. Materials and Methods

### 2.1. Mice

The *Lamc2^jeb^* mice are originally from Jackson Laboratory, strain #025467. The mice’s autosomal recessive mutation was due to a murine leukemia virus long terminal repeat insertion in *Lamc2* and exists on a C57BL/6J background [6,7]. All mice used in the following experiments were female littermates from the same generation, sired by the same male mouse, and the dams were littermates as well. Jefferson’s Institutional Animal Care & Use Committee (IACUC) provided approval for these mice experiments (protocol number 02141).

### 2.2. Mice Implantation of Trametinib

Mice received Trametinib (MedChemExpress, Monmouth Junction, NJ, USA, HY-10999) at 1 mg/kg/day through a micro-osmotic pump (Alzet, Cupertino, USA, Model 1004) at 10 weeks of age. The pumps deliver Trametinib at 0.11 µL per hour for 28 days, after which we had the pump surgically removed 42 days post-implant. Mice were harvested at 18 weeks of age or if they showed symptoms requiring ethical euthanasia. Micro-osmotic pumps were filled, weighed, and incubated in proper conditions according to the company’s (Alzet) instructions.

### 2.3. Mice Water Treatment with Losartan

Losartan (MedChemExpress, HY-17512) was administered in drinking water at 0.6 g/L concentration, starting at 10 weeks of age for 8 weeks. Mice were harvested at 18 weeks of age or if they showed symptoms requiring ethical euthanasia.

### 2.4. Severity Scoring

A blinded scoring was performed at the conclusion of the experiment by three individuals who were not involved or familiar with the JEB mice to establish trauma severity on the ears. The scale was from 1 to 5, with 1 being no severity and a representative image of undamaged ears shown and 5 was the most severe, with a representative image of late-stage mouse ear blistering shown from Sproule et al.’s paper’s Figure 3C [7].

### 2.5. Immunohistochemistry (IHC)

Immunohistochemistry was performed using the ABC-HRP protocol. All heat-induced epitope retrieval was performed with citrate buffer (pH = 6.0) except for CD4 staining, which used Tris-EDTA Buffer (pH = 9.0) and a methyl green counter stain. Please refer to Table A1 for a list of all antibodies used for IHC. All images were uploaded using Aperio CS2 ScanScope (Leica, Wetzlar, Germany).

### 2.6. Aperio Staining Analysis

We analyzed all immunohistochemistry images with ImageScope’s Positive Pixel Count 2004-08-11 algorithm, using the Noise to Signal Ratio (NSR) data from the annotations as quantification values. NSR is the ratio of strong-positive pixels to total pixels. Annotations included areas of the epidermis (from the stratum granulosum to the stratum basale) and dermis (both papillary and reticular layers). Blood vessels, hair follicles, keratin pearls, nerve endings, adipose tissue, and glands were excluded from the annotations.

### 2.7. Hematoxylin and Eosin (H&E) Staining

Our H&E staining used a Weigert’s Iron Hematoxylin Set (Sigma-Aldrich, St. Louis, MO, USA, HT1079) and Eosin Y (Fisher, Hampton, VA, USA, E-511). Then, 20X images were taken using the EVOS FL Auto Imaging system (Thermo Fisher Scientific, Hampton, VA, USA).

### 2.8. Statistics

Significance was determined by GraphPad Prism using an ordinary one-way ANOVA if the normality assumption was met, or Kruskal–Wallis test if otherwise. Specific multiple comparison corrections are specified next to each test in the figure legends. *p* ≤ 0.05 was considered significant and represented with a *, *p* ≤ 0.01 was represented with **, *p* ≤ 0.001 was represented with *** and *p* ≤ 0.0001 was represented with ****.

## 3. Results

### 3.1. Trametinib Had a Wide Range of Effect on Lamc2^jeb^ Mice Ear Damage

We investigated the effect of Trametinib both alone and with Losartan in a combination treatment in *Lamc2^jeb^* mice. Treatment began at 10 weeks of age and mice were sacrificed up to 8 weeks later (Figure 1A). Control mice received untreated water while Trametinib-treated mice received water with or without Losartan. Compared to control mice Trametinib treatment accelerated disease onset from 117 days (16.7 weeks) to 82 days (11.7 weeks) which was ameliorated in part by the addition of Losartan to 110 days (15.7 weeks, Figure 1B). These observations are independent of previously described sex bias because all mice in the study were female.

At the end of the experiment, all animals were sacrificed, photographed, and tissues were harvested for histological assessment. Disease severity was determined by the extent of tissue damage to the ear using a blinded severity score of photographic images. This analysis revealed a wide range of responses to Trametinib, with some mouse ears looking relatively unharmed while others were severely damaged with an increase in severity beyond control (Figure 1C). Out of the five Trametinib-treated mice, one mouse was euthanized at 16 weeks due to ethical concerns from pain-related behavior. Otherwise, the other four Trametinib-treated mice exhibited no change in pain-related behavior over the course of the experiment. The Trametinib and Losartan-treated mice, however, all responded similarly to control mice.

### 3.2. Severe Reactions to Trametinib Contribute to a Thinner Epidermis

Trametinib-treated mice experience a wide range in ear severity scores (Figure 1), and while the epidermal thickness is significantly reduced in all treated animals (Figure 2A), those mice with poor tolerance experience a thinner epidermis compared to untreated control (Figure 2B). Losartan co-treatment with Trametinib retains the same epidermal thickness as the untreated control, further validating that Losartan ameliorates Trametinib’s effect on inflamed murine skin (Figure 2C).

### 3.3. Severe Reactions to Trametinib Experience More Fibrosis with a Non-Significant Reduction in CD4 Expression

Out of five mice receiving Trametinib alone, three mice received a severity score rating above 3/5, which acted as the dividing score between a well-tolerated response to Trametinib and a severe reaction to Trametinib. One marker of fibrosis, αSMA, was measured using immunohistochemistry on *Lamc2^jeb^* ear skin. We noticed a significantly increased expression of αSMA in the severe response Trametinib mice compared to the control (Figure 3A), which concurs with our severity score rating given the level of blistering damage and granulation tissue that was observed macroscopically. While we did not see any significant change in CD4 expression in either Trametinib response group (Figure 3B), the severe response mice had a similarly low level of CD4 expression among their cohort. We then looked at the correlation of the severity score ratings of each individual Trametinib-treated mouse against the fold change values of αSMA and CD4 and noticed that with increasing ear damage severity, αSMA increased and CD4 decreased (Figure 3C).

### 3.4. Losartan Ameliorates Trametinib’s Fibrotic Effect

Four mice received Trametinib treatment in addition to Losartan throughout the course of the experiment, and they all exhibited a similarly phenotypic response. The increase in αSMA observed in the animals with a severe response to Trametinib was not apparent with the addition of Losartan (Figure 4A). While there was no change in CD4 expression when compared to control or severe response Trametinib mice (Figure 4B), there is a gradual increase in CD4 expression as the severity score of the Trametinib and Losartan-treated mice also increases (Figure 4C).

## 4. Discussion

There are few therapeutic options for patients with JEB, who live their lives with deteriorating blisters and increasing disease burden with age. To assess potential therapies to address or slow the increasing burden of disease we investigated the potential of repurposing Trametinib as an anti-fibrotic and anti-inflammatory compound in a preliminary in vivo study using a mouse model of JEB. We found that over half of the mice treated with Trametinib experience a more severe phenotype, marked by a thinner epidermis, an increase in fibrotic marker αSMA, and a non-significant decrease in the CD4 T cell infiltrate. In a combination treatment of Losartan and Trametinib, however, we found that Losartan ameliorated Trametinib’s adverse effects and restored epidermal thickness to the control phenotype, decreased αSMA expression compared to Trametinib-only treatment, and CD4 expression steadily increasing in response to greater ear damage severity. Overall, we find that Trametinib is not a viable therapeutic option for JEB, but Losartan is able to modulate some of Trametinib’s effects, potentially opening new lines of investigation to consider in terms of Losartan’s role in inflammatory conditions in EB research. Future work will assess Losartan in monotherapy in this mouse model.

The wide range in effect among littermate JEB mice with Trametinib treatment was an unpredicted phenomenon and one that prevented the expansion of our treatment cohorts since accelerated disease phenotype presented ethical issues that prevented continued study.

The JEB mouse model we used is an inbred strain provided by Jackson Labs on the C57BL/6 background, and we controlled the breeding by crossing littermates for at least twenty generations. Inbred strains are reported to have impaired homeostatic mechanisms, and therefore are more affected by drug or hormone administration compared to hybrid mice [28]. One example of this variability in drug sensitivity in inbred populations is a study performed using the anti-psychotic chlorpromazine, in which adult mice of both sexes from strains such as C57BL/6 had their activity levels tested post-treatment [29]. Even at higher doses of chlorpromazine, only 90% of C57BL/6 mice experienced physiological effects from the drug. The genetic mechanism behind this observation is still poorly understood [30], and we observe similar differences in response in our study. T cell phenotypes are also affected among inbred mice and may be influencing our observations here [31].

Trametinib as a chemotherapeutic has also demonstrated a wide range of effects among human patients in clinical trials, either as a monotherapy or as a co-treatment [32]. Out of over 200 patients with advanced solid tumors enrolled in a phase 1 study, only 10% of patients responded to Trametinib based on clinical activity observations [33]. Even in cases of cutaneous cancers such as melanoma, Trametinib achieved a 40% response rate among 97 patients with melanoma [34]. In a phase 1 trial evaluating Trametinib co-treatment with another anticancer agent on 47 small-cell lung carcinoma-afflicted patients, 21% responded at least partially to the combination treatment [35]. Of course, the obvious difference between our study and the listed trials besides the patients being human is that we attempted to use Trametinib as a fibrosis preventative rather than treating established and advanced cancer, an inherently heterogeneous disease. Trametinib’s effects in our model provide a unique outlook and uncover limitations on previously unstudied uses for Trametinib.

One obvious consideration for the variability is the mode of delivery; the osmotic pumps, since we did not measure levels of Trametinib in the treated animals over time it is conceivable that the two treated mice with mild response did not receive a similar dose due to issues with the pump. However, the Trametinib and Losartan-treated mice also all received pumps with Trametinib from the same lot, and we did not see any variability among these mice. We do note that those mice with a severe reaction did have a thinner epidermis (Figure 1) compared with mice without a severe reaction, and this could be an indicator of drug exposure since Trametinib targets MEK with the potential to reduce proliferation. One possible reason for the variability could be environmental and related to ear grooming triggering a cycle of inflammation, itch, and disease onset, which is exacerbated by Trametinib’s effects on proliferation. Other possible reasons for this variability include drug homeostasis inconsistency among inbred mice (potentially influenced by genetic drift or epigenetic phenomena) and Trametinib’s overall response rate.

The effect of Trametinib on epidermal thickness is intriguing, and logic would dictate that inhibition of MEK in a highly proliferative tissue such as the epidermis would lead to a reduction in cellular content (in this case keratinocytes) and subsequent thickening. In the context of JEB, it is known that human patients have a stem cell defect, with laminin 332 contributing to stem cell maintenance [36,37], and this may well be compounded through the inhibition of proliferation. Indeed, it has been demonstrated that Trametinib inhibits dermal stem cells as well [38]. Losartan on the other hand has demonstrated the ability to improve stem cell niches derived from adipose tissue and muscle [39,40], which provides another potential mechanism of action to explore in epidermal stem cells. Furthermore, and as mentioned earlier, the effects could be accelerated by environmental triggers leading to a worsening of the disease phenotype in the context of Trametinib treatment.

In addition to CD4, other markers of adaptive immunity we investigated include CD3, CD8, and CD45 [41]. CD3 is a central T cell marker in that it appears at all stages of T cell development, and its complex with T cell receptors (CD3-TCR) is important in recognizing antigens for the activation of T lymphocytes and mediating antigen-specific immune responses [42]. On the other hand, CD8+ lymphocytes, or cytotoxic T cells, facilitate adaptive immunity with CD4+ lymphocytes. Together, these cells are the main constituents of T cell-mediated responses since CD8 cells are known to fight viral infections and cancer cells while CD4 cells support CD8 cell function [43]. Lastly, CD45 is a JAK phosphatase that controls kinase dephosphorylation for antigen receptor signaling and is expressed on all nucleated hematopoietic cells, making it a more general marker of study in inflammation [44]. These markers are all interrelated in that CD4, CD8, and CD45 are responsible for the regulation of the CD3-TCR complex, which activates the tyrosine kinase signaling [45].

We found no changes in CD3, CD8, or CD45 with monotherapy Trametinib treatment but CD8 was significantly inhibited in Trametinib and Losartan co-treatment (Appendix A). Previous literature has found that inhibiting MEK, Trametinib’s mechanism of action, leads to suppression of T cells presumably through a reduction in proliferation in murine models of ovarian, breast, and colorectal cancer [46,47]. Here, we are seeing a non-significant reduction in immune cells with Trametinib treatment indicating that epidermal infiltration in the context of JEB is not as widely affected. It is possible that the severe reactions to Trametinib are due to an influx of cytokines that induce negative feedback inhibition on CD4+ lymphocytes [48,49,50,51] that would normally contribute to a greater circulation of cytokines to facilitate homeostasis in wound repair. Such aberrant inflammation from cytokines in the wound environment is compounded and leads to greater fibrosis, as suggested by the αSMA-increased trends. A larger study capable of stratifying severity within the Trametinib arm would be required to determine whether this hypothesis is true; however, the overall observation is that Trametinib has the potential to accelerate disease phenotype in the JEB model and as such (for ethical reasons) we have not pursued this question further.

Losartan is a well-studied repurposed compound in the field of epidermolysis bullosa, especially for the subtype of dystrophic EB. Losartan has been shown to reduce inflammation in other diseases such as HIV but has never demonstrated the ability to target CD4+ lymphocytes [52,53]. Losartan’s class of compound, angiotensin receptor blockers, exert anti-inflammatory effects generally as well [54]. Since Losartan has proven to mitigate cytokines such as Transforming Growth Factor-β (TGFβ) [55], especially in areas of recurrent blistering, which thereby also downregulate αSMA+ fibroblasts [27], Trametinib’s suppression of CD4 is not compounded by the presence of other cytokines. This potential mechanism thereby allows the wound to heal more progressively in the last four weeks of treatment with just Losartan (Figure 5). While Losartan in conjunction with Trametinib could not improve the JEB phenotype compared to control, our studies suggest a potential novel mechanism of action to be further elucidated for both compounds.

## 5. Conclusions

Junctional epidermolysis bullosa is a skin blistering disease with cycles of wounding and chronic inflammation as a chief medical concern associated with the disease. We worked with a mouse model of JEB involving a mutation in laminin-332, with the goal of determining the potential to repurpose anti-fibrotic compounds such as Trametinib, either in monotherapy or in combination with Losartan. We found that Trametinib monotherapy worsened the clinical phenotype in the majority of mice, with epidermal ear thinning in those mice with more severe disease. Of those mice that experienced severe symptoms and marked by an increase in αSMA (marker of activated fibroblasts), an inverse correlation with decreasing CD4+ T cells was observed. When we combined Trametinib and Losartan together as a co-treatment Trametinib’s detrimental effects were mitigated, especially the epidermal thinning, but the clinical phenotype of the mice was not completely ameliorated compared to untreated control. We also noted a significant decrease in CD8+ T cells with Losartan co-treatment, and that despite ear damage severity there was no inverse correlation trend in CD4+ T cell expression with αSMA that we see with Trametinib monotherapy. Future directions include repurposing Losartan monotherapy in the JEB mouse model, as well as identifying other compounds that could work synergistically with Losartan. While we need to exclude Trametinib as a potential treatment for JEB due to the compromised structural integrity of the skin and upregulated fibrosis, our pilot study offered additional insight into the immunological effects of this popular chemotherapy drug.

## Figures and Tables

**Figure 1 biomolecules-13-00740-f001:**
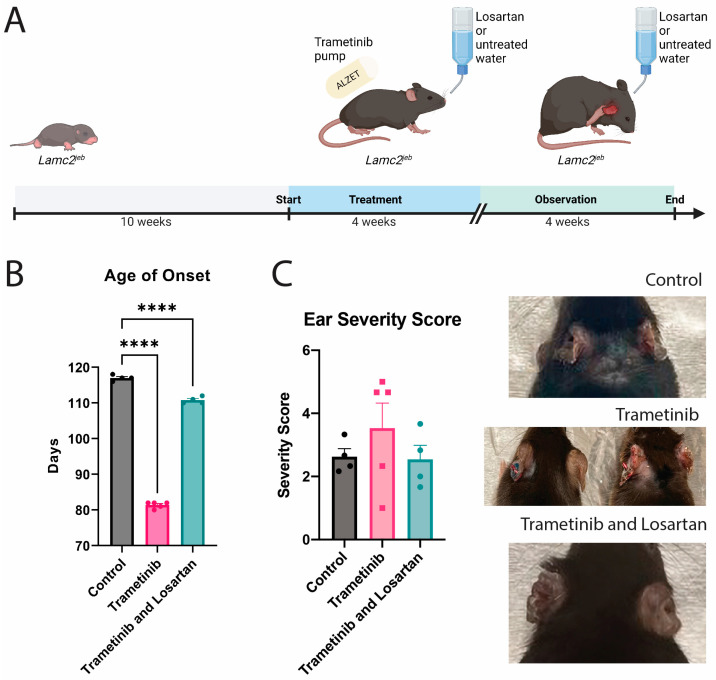
Trametinib accelerated JEB disease onset. (**A**) Schematic showing experiment timeline in *Lamc2^jeb^* mice. Trametinib was delivered via an infusion pump starting at 10 weeks old for four weeks with or without Losartan-treated water. Mice were observed with or without Losartan water for the following four weeks before harvesting; (**B**) Age of phenotype onset graph for untreated control, Trametinib, and Trametinib and Losartan treated mice. Age of onset is defined as the day when evidence of disease phenotype is observed, i.e., redness, crusting, or blistering. One-way ANOVA (Dunnett correction) was performed, with both treatment groups significantly accelerating phenotype onset (****: *p* ≤ 0.0001); (**C**) Ear severity score means + SEM. One-way ANOVA (Dunnett correction) was performed, with treated groups found to have no significant difference from control (*p* ≥ 0.05). Representative images are shown of the mice at harvesting, with untreated control mice on top. Trametinib-treated images in the middle show both well-tolerated (left) and severe reaction (right) mice ears. Bottom image shows a representative image of a Trametinib and Losartan-treated mouse ears.

**Figure 2 biomolecules-13-00740-f002:**
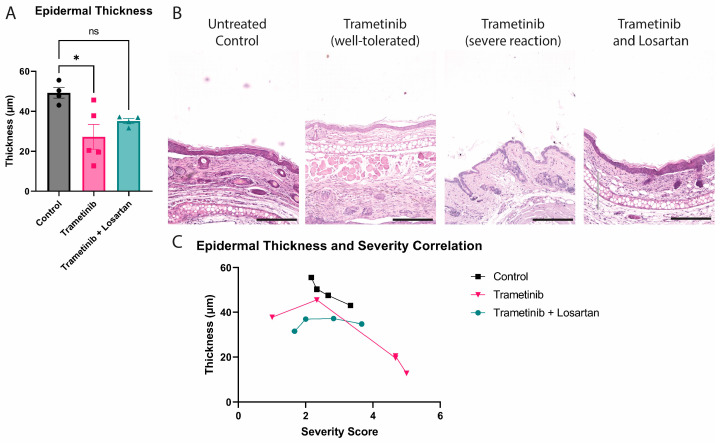
Trametinib reduces epidermal thickness of JEB mice. (**A**) Graph showing mean ± SEM of epidermal thickness with one-way ANOVA analysis (Dunnett correction) performed, with Trametinib significantly decreased (*: *p* ≤ 0.05, ns: *p* ≥ 0.05); (**B**) Representative images of H&E stainings of the ears from untreated control, well-tolerated Trametinib-treated mice, severe reaction Trametinib-treated mice, and Trametinib and Losartan-treated mice; (**C**) XY-plot graphing severity score ratings (X) of individual mice by the epidermal thickness (Y). Black line represents control, pink line represents Trametinib-treated mice, and green line represents Trametinib and Losartan-treated mice.

**Figure 3 biomolecules-13-00740-f003:**
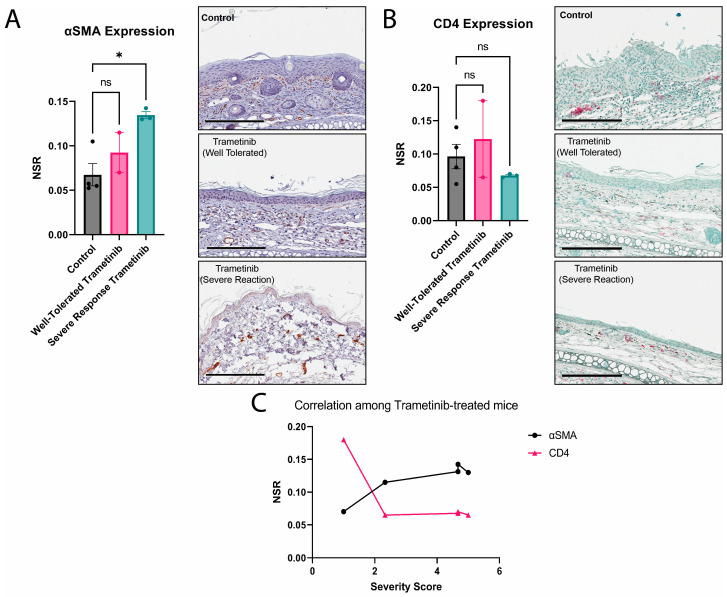
*Lamc2^jeb^* mice experience increased αSMA with poor tolerance of Trametinib. (**A**) Graph showing mean ± SEM of αSMA expression with Kruskal–Wallis test performed (Dunn’s correction) (left) and representative images of IHC staining (right). (*: *p* ≤ 0.05, ns: *p* ≥ 0.05); (**B**) Graph showing mean ± SEM of CD4 expression with one-way ANOVA analysis (Dunnett correction) performed (left) and representative images of IHC staining (right). (ns: *p* ≥ 0.05); (**C**) XY-plot graphing severity score ratings (X) of individual mice by the NSR of IHC markers (Y). Black line represents αSMA NSR values while pink line represents CD4 NSR values.

**Figure 4 biomolecules-13-00740-f004:**
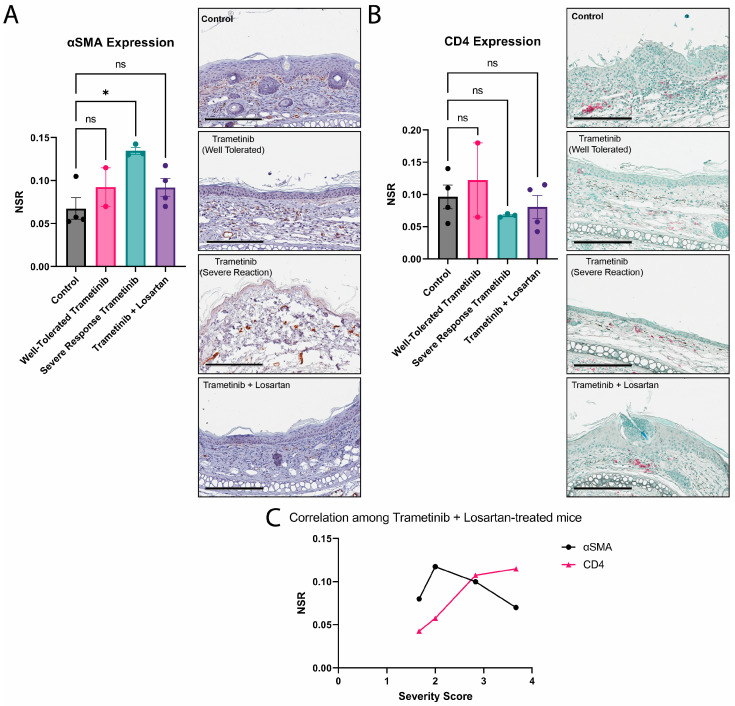
Losartan upregulates CD4 expression in *Lamc2^jeb^* mice when treated with Trametinib. (**A**) Graph showing mean ± SEM of αSMA expression with Kruskal–Wallis test (Dunn’s correction) performed (left) and representative images of IHC staining (right). (*: *p* ≤ 0.05, ns: *p* ≥ 0.05); (**B**) Graph showing mean ± SEM of CD4 expression with one-way ANOVA analysis (Holm-Šídák correction) performed (left) and representative images of IHC staining (right). (*: *p* ≤ 0.05, ns: *p* ≥ 0.05); (**C**) XY-plot graphing severity score ratings (X) of individual mice by the NSR of IHC markers (Y). Black line represents αSMA NSR values while pink line represents CD4 NSR values.

**Figure 5 biomolecules-13-00740-f005:**
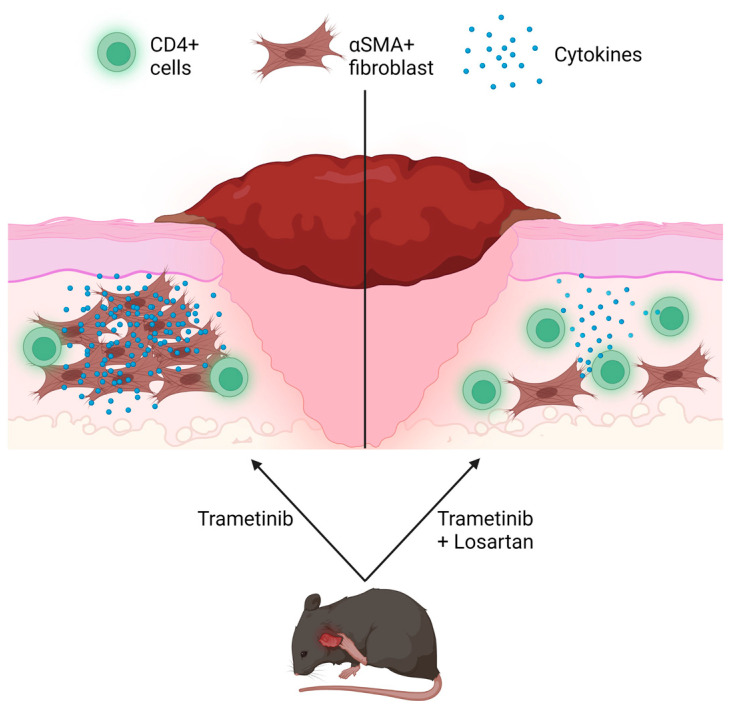
Losartan potentially ameliorates Trametinib’s inflammatory effect. A graphical illustration comparing the difference between Trametinib and Trametinib with Losartan treatment in *Lamc2^jeb^* mice. With Trametinib alone, CD4 expression decreases while αSMA+ fibroblasts increase. Losartan in addition to Trametinib leads to similar expressions of CD4 and αSMA as control mice, possibly due to less cytokines circulating in the wound bed such as TGFβ. Made with Biorender.com.

## Data Availability

All data are presented within the manuscript.

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
