# Peer review of "Trametinib-Induced Epidermal Thinning Accelerates a Mouse Model of Junctional Epidermolysis Bullosa"

_biomolecules, 2023, doi:10.3390/biom13050740_

Round 1
Reviewer 1 Report
Tartaglia et al. report on a small preclinical study of potential anti-fibrotic agents (MEK inhibitor, trametinib, +/- angiotensin receptor blocker, losartan) in a murine model of JEB, comparing clinical severity of ear wounds, epidermal thickness, immune infiltrate and fibrotic marker (alpha-SMA) by IHC, and H&E. Trametinib alone surprisingly accelerated clinical manifestations of JEB compared to a control group, while combined administration of losartan with trametinib seemed to ameliorate this accelerated phenotype. While only 5 animals in the trametinib alone group, 3 had a severe phenotype with epidermal thinning, increased alpha-SMA staining and non-significant decrease in CD4 T cell infiltrate. Findings led researchers to halt further investigation of trametinib though mechanistic implications are interesting. Losartan, which has been investigated both pre-clinically and clinically, in another form of EB, recessive dystrophic EB, may be beneficial in JEB as well.
The manuscript is generally well written and easy to read with important findings en route to a targeted therapy to improve JEB wound healing.
Comments:
Major concerns:
(1) The group sizes are very small (control n=4, trametinib alone n=5, trametinib + losartan n=4) limiting confidence in conclusions. More importantly, a control group of losartan alone is missing.
(2) The discussion lacks hypotheses to explain the varied responses to MEK inhibition (disease severity, pain, etc). Were there any concerns with the drug infusion by implanted pump? Were devices all empty upon removal? Could drug have been infused faster than anticipated in an inconsistent manner?
Minor concerns:
(3) Line 59: The statement beginning "One such T cell marker" suggests that the surface protein CD4 itself contributes to fibrosis. I believe the authors are wanting to indicate CD4 as a marker of an infiltrating T cell population that contributes to fibrosis (as they go on to describe)
(4) Line 63: Do the author mean "secreting" instead of "circulating"?
(5) Line 100: What was the frequency of severity scoring? Weekly?
(6) Line 113: NSR undefined. Presumably "noise-to-signal ratio"
(7) Line 119: Statistics section, any corrections for multiple comparisons used?
(8) Line 215: "non-significant decrease in the T cell marker CD4" - the marker itself isn't clinically relevant, would recommend rephrasing to "non-significant decrease in the CD4 T cell infiltrate."
(9) Figure 1 title: Would recommend changing "increased" to "accelerated"
(10) Figure 1A: The current figure shows all patients receiving losartan from 4-8 wks of age.
(11) Figure 1B: Is "age of onset" defined at the first time point a severity score of 2 is noted?
(12) Between Fig 1A and 1B (and in the text), would be helpful to have timeline consistency, either reporting both in weeks or days
Author Response
Reviewer 1:
- The group sizes are very small (control n=4, trametinib alone n=5, trametinib + losartan n=4) limiting confidence in conclusions. More importantly, a control group of losartan alone is missing.
Response: We thank the reviewer for reading the manuscript thoroughly and for their insightful comments. We acknowledge the small size of the treatment groups as this was a preliminary study of the potential to repurpose Trametinib for JEB. We did not increase the sample size due to ethical concerns upon seeing the severe reaction to Trametinib treatment in our first cohort of 4-5 mice.
We did have a control group of Losartan-treated mice (5 mice total) but did not include them in our manuscript due to the described sex difference in age of onset of the phenotype and that the mice in this group were male while the untreated, Trametinib, and Trametinib + Losartan-treated mice were all female. This dichotomy was due to our experimental plan requiring staggering of the experiments because our colony size and cage number were limited, and we wanted to keep experimental arms limited to sex-matched littermates from the same mother. We were planning to include female mice for the losartan treatment and male mice for the Trametinib treatment in the second round of experiments but, as noted above, we did not pursue this due to the ethical concerns regarding Trametinib treatment. Rather than discard the data from this study we seek to publish the observation that Trametinib accelerates disease severity in this manuscript, albeit as a preliminary study and albeit without the Losartan only treatment group. The sex difference in the age of onset of the phenotype has been previously noted: Sproule et al published findings indicating that there is a sex bias among the JEB mice in terms of disease phenotype onset (male mice develop disease earlier than female mice), but studies regarding differences in severity have not been performed[1]. Thus, in order to negate the confounding factor of sex, we excluded the Losartan-treated mice (which had accelerated disease onset as expected for male mice compared to untreated female control but no difference in ear severity scoring). We do plan to initiate experiments testing Losartan only in the future under better-controlled conditions.
- The discussion lacks hypotheses to explain the varied responses to MEK inhibition (disease severity, pain, etc). Were there any concerns with the drug infusion by implanted pump? Were devices all empty upon removal? Could drug have been infused faster than anticipated in an inconsistent manner?
Response: The osmotic pumps we used are commercially available pumps from Alzet with a quality-tested 0.11μL/hour flow rate for 672 hours (28 days). We followed all protocol instructions from the company that included overfilling the pumps slowly with liquid in order to prevent trapped air bubbles and submerging pumps in saline at 37ËšC for 48 hours prior to surgery to ensure pumps infuse at a constant rate once implanted. Since the pumps act via osmosis the pumps contained fluid at day 42, the latest day the company recommends leaving pumps safely in mice, but we did not measure the amount of Trametinib in the remaining fluid and nor did we assess the pharmacokinetics of Trametinib in treated animals.
Whilst we are confident in the mice all being treated with Trametinib, it is possible one or more pumps were faulty and infused at an inconsistent rate. We did not measure serum levels of Trametinib in the mice during the experiment since these assays are not readily available. We have added a paragraph in our discussion addressing potential hypotheses for the variable response to Trametinib among JEB littermates, thank you for this suggestion.
- Line 59: The statement beginning "One such T cell marker" suggests that the surface protein CD4 itself contributes to fibrosis. I believe the authors are wanting to indicate CD4 as a marker of an infiltrating T cell population that contributes to fibrosis (as they go on to describe)
Response: We thank the reviewer for pointing out this vague statement and have corrected the sentence to “One marker of infiltrating T cells that contributes to fibrosis is cluster of differentiation 4 (CD4)…”
- Line 63: Do the author mean "secreting" instead of "circulating"?
Response: We thank the reviewer for identifying this unclear wording and have corrected the phrase to “CD4+ T cells … by secreting inflammatory cytokines…”
- Line 100: What was the frequency of severity scoring? Weekly?
Response: Severity scoring was performed at the end of the experiment based on photographs of unlabeled mice. We clarified the time of severity scoring in the methods section.
- Line 113: NSR undefined. Presumably "noise-to-signal ratio"
Response: We thank the reviewer for identifying our lack of clarity in this abbreviation, and have clarified that NSR stands for noise to signal ratio in the methods section.
- Line 119: Statistics section, any corrections for multiple comparisons used?
Response: We thank the reviewer for identifying our lack of detail in statistical tests. All tests were performed using Graphpad Prism with Multiple Comparisons’ recommended corrections. Individual test corrections are now specified in every figure legend.
- Line 215: "non-significant decrease in the T cell marker CD4" - the marker itself isn't clinically relevant, would recommend rephrasing to "non-significant decrease in the CD4 T cell infiltrate."
Response: We thank the reviewer for this recommended clarifier and have changed the wording accordingly.
- Figure 1 title: Would recommend changing "increased" to "accelerated"
Response: We thank the reviewer for this helpful title adjustment and have changed the wording accordingly.
- Figure 1A: The current figure shows all patients receiving losartan from 4-8 wks of age.
Response: We thank the reviewer for this helpful advice and have adjusted the wording accordingly.
- Figure 1B: Is "age of onset" defined at the first time point a severity score of 2 is noted?
Response: We thank the reviewer for pointing out this unclear detail and have clarified in line 157; age of onset is evidence of disease as indicated by redness, crusting or blistering of the ear.
- Between Fig 1A and 1B (and in the text), would be helpful to have timeline consistency, either reporting both in weeks or days
Response: We have included age of onset in weeks in parentheses next to the days (lines 130-131).
References
- Sproule, T.J.; Bubier, J.A.; Grandi, F.C.; Sun, V.Z.; Philip, V.M.; McPhee, C.G.; Adkins, E.B.; Sundberg, J.P.; Roopenian, D.C. Molecular identification of collagen 17a1 as a major genetic modifier of laminin gamma 2 mutation-induced junctional epidermolysis bullosa in mice. PLoS Genet 2014, 10, e1004068, doi:10.1371/journal.pgen.1004068.
Reviewer 2 Report
Grace et al. reported that Trametinib accelerated epidermal thinning in a mouse model of JEB, which was rescued when combined with Losartan.
The study was well and appropriately analyzed. The reviewer felt that the study would be even better if there were genetic differences and epigenetic analysis between the groups that had a severe reaction to Trametinib and those that did not.
Author Response
Reviewer 2:
The study was well and appropriately analyzed. The reviewer felt that the study would be even better if there were genetic differences and epigenetic analysis between the groups that had a severe reaction to Trametinib and those that did not.
Response: We thank the reviewer for reading our manuscript and appreciating our findings. As for genetic differences we acknowledge that there could be a very small amount of genetic drift in our homozygous, inbred colony of genetically identical mice, but we feel that this would not contribute to such a difference in response. Likewise, epigenetic changes would be unlikely but may be influenced by environment and behavior, such as excesive grooming. We included a paragraph in the discussion addressing potential reasons for Trametinib having a wide range in effect among JEB female littermates; published studies have noted that drug response, even in genetically identical inbred mice, does indeed vary and that this depends on strain and could be due to a number of factors[2-4], which in the case of skin fragility and inflammation may be environmentally driven and exacerbated by grooming.
References
- Chai, C.K. Response of inbred and F1 hybrid mice to hormone. Nature 1960, 185, 514-518, doi:10.1038/185514a0.
- Plotnikoff, N. Drug Resistance due to Inbreeding. Science 1961, 134, 1881-1882, doi:doi:10.1126/science.134.3493.1881.
- Vaickus, L.J.; Bouchard, J.; Kim, J.; Natarajan, S.; Remick, D.G. Inbred and outbred mice have equivalent variability in a cockroach allergen-induced model of asthma. Comp Med 2010, 60, 420-426.
Round 2
Reviewer 1 Report
The authors' responses to reviewer concerns and queries were well considered and edits valuable additions to the manuscript. No further concerns were identified.